# Exercise-Induced Oxidative Stress, Nitric Oxide and Plasma Amino Acid Profile in Recreational Runners with Vegetarian and Non-Vegetarian Dietary Patterns

**DOI:** 10.3390/nu11081875

**Published:** 2019-08-13

**Authors:** Josefine Nebl, Kathrin Drabert, Sven Haufe, Paulina Wasserfurth, Julian Eigendorf, Uwe Tegtbur, Andreas Hahn, Dimitrios Tsikas

**Affiliations:** 1Institute of Food Science and Human Nutrition, Leibniz University Hannover, 30159 Hannover, Germany; 2Institute of Toxicology, Hannover Medical School, 30625 Hannover, Germany; 3Institute of Sports Medicine, Hannover Medical School, 30625 Hannover, Germany

**Keywords:** diet, exercise, malondialdehyde, plasma, nitric oxide, vegan, vegetarian

## Abstract

This study investigated the exercise-induced changes in oxidative stress, nitric oxide (NO) metabolism and amino acid profile in plasma of omnivorous (OMN, *n* = 25), lacto-ovo-vegetarian (LOV, *n* = 25) and vegan (VEG, *n* = 23) recreational runners. Oxidative stress was measured as malondialdehyde (MDA), NO as nitrite and nitrate, and various amino acids, including homoarginine and guanidinoacetate, the precursor of creatine. All analytes were measured by validated stable-isotope dilution gas chromatographic-mass spectrometric methods. Pre-exercise, VEG had the highest MDA and nitrate concentrations, whereas nitrite concentration was highest in LOV. Amino acid profiles differed between the groups, with guanidinoacetate being highest in OMN. Upon acute exercise, MDA increased in the LOV and VEG group, whereas nitrate, nitrite and creatinine did not change. Amino acid profiles changed post-exercise in all groups, with the greatest changes being observed for alanine (+28% in OMN, +21% in LOV and +28% in VEG). Pre-exercise, OMN, LOV and VEG recreational runners differ with respect to oxidative stress, NO metabolism and amino acid profiles, in part due to their different dietary pattern. Exercise elicited different changes in oxidative stress with no changes in NO metabolism and closely comparable elevations in alanine. Guanidinoacetate seems to be differently utilized in OMN, LOV and VEG, pre- and post-exercise.

## 1. Introduction

Intense exercise induces oxidative stress. The majority of the reactive oxygen and nitrogen species including free radicals cannot be determined in biological samples such as blood, because of their high chemical reactivity, their low concentration and extremely short half-life. Instead of, their stable and analytically accessible metabolites are measured. Biological malondialdehyde (MDA) is a product of lipid-peroxidation and one of the most widely used and generally accepted biomarkers of oxidative stress [1]. Numerous human studies revealed associations between oxidative stress and the pathogenesis of atherosclerosis, endothelial dysfunction and cardiovascular diseases [2,3,4,5]. On the other hand, however, oxidative stress has been associated with various signalling pathways including nitric oxide (NO). It is assumed that oxidative stress may exert health-promoting effects on physical activity and, consequently, on training adaptations and human physiology, such as insulin sensitivity [6,7]. For these reasons, the equivocal role of exercise-induced oxidative stress needs further clarification [8].

In addition to the endogenous defence systems, dietary antioxidant intake (e.g., vitamin C and vitamin E) is crucial to avoid adverse effects of oxidative stress. Antioxidant supplements are widely consumed in everyday life and in sports. In general, a balanced mixed diet can provide the athlete with adequate amounts of antioxidants. However, dietary intake of antioxidants in athletes is often insufficient [9,10]. Plant-based diets, such as those consumed in vegetarianism and veganism, are rich in antioxidants and are assumed to prevent from oxidative damage. In fact, plant-based diets seem to be advantageous over average mixed diets in various diseases, such as diabetes, hypertension and cancer [11,12,13]. Several groups reported on positive effects of plant-based diets on exercise-induced oxidative stress and inflammatory parameters [14,15]. However, these data are based solely on the assumption that vegetarians have a higher intake of antioxidants compared to omnivores.

A recent study investigated the oxidative status in moderately active vegan, vegetarian and omnivorous men [16]. In that study, a higher antioxidant capacity was measured in omnivores compared to both vegans and vegetarians. This in vitro study found that incubation of the rat cardiac myoblastic cell line H9c2 with vegan serum samples elevated thiobarbituric acid-reactive substances and induced cell death [16]. Yet, it is unclear whether oxidative burden depends on the dietary pattern under physical stress in vivo. Thus far, there are no studies to demonstrate that the assumed increased intake of antioxidants indeed provides protection against exercise-induced oxidative stress [15].

Amino acids undergo tightly controlled homeostatic regulation. Their concentrations in blood reflect a balanced interaction between dietary intake, endogenous synthesis, catabolic and anabolic processes [17,18]. Exercise-induced adaptations have been observed in amino acid metabolism [19]. Long-lasting endurance events showed an overall decrease by 15–30% in total circulating amino acid concentration [20,21,22], while the concentration of aromatic amino acids often increased by 6–11% [23,24]. Results of previous studies show that the plasma amino acid profile differs between male omnivorous, vegetarian and vegan non-athletes [25,26,27,28,29,30,31]. However, there are no such data from athletes. Additionally, no data on exercise-induced changes in circulating amino acids and their metabolites have been reported thus far.

We have hypothesized that there are differences between omnivorous, lacto-ovo-vegetarian and vegan recreational runners with respect to exercise-induced oxidative stress, NO metabolism and circulating amino acids. The present study was conducted to prove this hypothesis by determining the concentrations of circulating MDA as a measure of lipid peroxidation, nitrite and nitrate as the major NO metabolites, and several amino acids and their metabolites in plasma samples collected in a previously reported study on healthy omnivorous, lacto-ovo-vegetarian and vegan recreational runners before and after laboratory physical exercise tests [32]. The flow chart of the study is reported as supplementary information (Appendix A). All analytes were measured by using fully validated and clinically proven gas chromatographic-mass spectrometric methods and stable-isotope labelled analogs as internal standards [33,34]. The method used for the measurement of the amino acids in the plasma samples provides the sum concentration for Ile and Leu (Ile+Leu), Asp and Asn (Asp+Asn), Glu and Gln (Glu+Gln), and for Orn and Cit (Orn+Cit). Thus, for these amino acid pairs their summed concentration is reported and considered in statistical analyses. The global arginine bioavailability ratio was calculated by dividing the plasma Arg concentration by the sum of the concentrations of Orn and Cit.

## 2. Materials and Methods

### 2.1. Study Design, Participants, Physical Exercise and Blood Sampling

The cross-sectional study on which the present work is based on has been previously reported in detail [32] (Appendix A). Subjects were recruited via advertisements from the general population in Hannover (Germany). The main inclusion criteria were: Omnivorous (OMN), lacto-ovo-vegetarian (LOV) or vegan (VEG) diet for at least half a year, body mass index (BMI) between 18.5 and 25.0 kg/m^2^ and regular running exercise two to five times per week. Exclusion criteria were: Any cardiovascular, metabolic or malignant disease, gastrointestinal diseases, pregnancy or lactation, nutrient intolerances, drug and alcohol dependency, concurrent participation in another clinical study, participation in a study in the last 30 days and retraction of the consent by the subject. Inclusion and exclusion criteria were queried using a screening questionnaire. Participants were matched according to age and gender. Omnivores consumed both plant and animal foods, lacto-ovo vegetarians excluded meat and fish from their diet and vegans only consumed plant-based foods. Food and beverages consumed in the last 24 h were recorded via 24 h dietary recall [32]. Seventy-six healthy, non-smoking OMN, LOV and VEG recreational runners conducted incremental 20–30 min lasting stress on a bicycle ergometer (Excalibur, Lode B.V., Groningen, Netherlands). Venous blood samples (7.5-mL EDTA monovettes, Sarstedt^®^, Nümbrecht, Germany) were collected from 73 subjects before and after the exercise load. No blood samples could be taken from three subjects, either before or/and after exercise test (Appendix A). There were no statistically significant differences regarding age, gender, BMI and training habit between the groups (Table 1). Dietary intake of antioxidants, amino acids and fatty acids by diet group is reported in Table 2.

Ethical approval was provided by the Ethics Committee at the Medical Chamber of Lower Saxony (Hannover, Germany; 12/2017). In accordance with the Declaration of Helsinki, written informed consent was obtained from all subjects prior to their participation in the study. This study is registered in the German Clinical Trial Register (DRKS00012377).

### 2.2. Sample Preparation and Biochemical Analyses

EDTA-anticoagulated blood was centrifuged (4 °C, 1620× *g*), 500 µL plasma aliquots were immediately aliquoted in 1.5 mL Eppendorf Tubes^®^ (Eppendorf AG, Hamburg, Germany) and frozen at –80 °C until analysis. The plasma samples were transferred frozen on dry-ice to the Institute of Toxicology at Hannover Medical School and stored in frozen state at −20 °C until analysis within the next few days. On each day of analysis, a certain number of plasma samples left thaw at room temperature. Aliquots of 100 µL of the thawed samples were taken for the simultaneous analysis of MDA, nitrite, nitrate and creatinine and transferred into 1.5 mL glass vials (Macherey-Nagel, Düren, Germany). For the analysis of amino acids, further 10 μL plasma aliquots were taken and transferred into 1.5 mL glass vials as well.

Plasma MDA, nitrate, nitrite and creatinine were analysed simultaneously as reported elsewhere [33]. Aliquots (10 µL) of a mixture of the internal standards containing 400 µM [^15^N]nitrate, 40 µM [^15^N]nitrite and 1 mM d_3_-creatinine in distilled water were added to the 100-µL plasma aliquots. Aliquots (15.4 µL) of the internal standard d_2_-MDA solution in 10 mM HCl (65 µM) were also added. The final concentrations of the stable-isotope labelled internal standards were 40 µM for [^15^N]nitrate, 4 µM for [^15^N]nitrite, 100 µM d_3_-creatinine and 10 µM d_2_-MDA with respect to the plasma volume. After addition of acetone (400 µL) and the derivatization reagent pentafluorobenzyl bromide (10 µL) to the samples, the glass vials were tightly closed and then heated for 60 min at 50 °C. After cooling to room temperature, acetone was removed under a gentle stream of nitrogen. Extraction of excess pentafluorobenzyl bromide and the reaction products from the remaining aqueous phase was carried out by adding ethyl acetate (1 mL) and by vortex-mixing the mixtures for two minutes at the highest speed using a Heidolph vortex mixer model Reax 2000 (Schwabach, Germany). After centrifugation (5 min, 3350× *g*) the upper organic phase was decanted and dried over anhydrous Na_2_SO_4_ (about 10 mg per sample). Subsequently, the samples were centrifuged again (5 min, 3350× *g*) and 750 µL aliquots of the organic phase were transferred into 1.8 mL autosampler glass vials (Macherey-Nagel; Düren, Germany) for gas chromatography-mass spectrometry analysis.

Plasma amino acids were analyzed simultaneously by gas chromatography-mass spectrometry in 10 µL aliquots using trideutero-methyl esters of the individual amino acids as the internal standards after preparation of the methyl ester pentafluoropropionyl derivatives as reported elsewhere [34].

### 2.3. Gas Chromatographic-Mass Spectrometric Analyses

Gas chromatographic-mass spectrometric analyses were performed on a single quadrupole mass spectrometer model ISQ directly interfaced with a Trace 1310 series gas chromatograph equipped with an autosampler AS 1310 from ThermoFisher (Dreieich, Germany). Different oven temperature programs were used for the separation of the derivatives of nitrate, nitrite, creatinine and MDA, on the one hand, and of the derivatives of the amino acids and their metabolites, on the other hand. Selected-ion monitoring of specific anions for unlabelled nitrate, nitrite, creatinine, MDA and for their stable-isotope labelled analogues was performed [33]. Amino acids were analysed by selected-ion monitoring of specific anions for endogenous and their stable-isotope labelled analogues as reported previously [34].

### 2.4. Data Analysis and Statistical Methods

Data are presented as mean ± standard deviation (SD). To control distribution, the Kolmogorov–Smirnov test was used. To evaluate differences between the three diet groups, a one-way analysis of variance was used for parametric data. For non-parametric data the Kruskal–Wallis test was performed. For statistically significant differences a post hoc test with Bonferroni correction was conducted to analyse differences between the individual groups. To examine differences between pre- and post-exercise within a group, the *t*-test (for parametric data) and the Mann–Whitney U test (for non-parametric data) were used. Further, to calculate correlations between parametric data, the Pearson correlation was computed. Finally, to assess associations between non-parametric data, Spearman’s rho correlation was used. Values of *p* ≤ 0.05 were regarded as statistically significant. All statistical analyses were conducted using SPSS software (IBM SPSS Statistics 24.0; Chicago, IL, USA) and GraphPad Prism 7.02 (GraphPad Software Inc., San Diego, CA, USA).

## 3. Results

The pre-exercise and post-exercise plasma concentrations of the biochemical markers measured in the present study are summarized in Table 3.

### 3.1. Pre-Exercise Concentrations

The highest pre-exercise plasma MDA (*p*_LOV_ = 0.020) and plasma nitrate (*p*_OMN_ = 0.049) concentrations were observed in VEG, suggesting higher pre-exercise oxidative stress and NO synthesis in this group. The highest pre-exercise plasma nitrite concentration was found in LOV (*p*_OMN_ < 0.001), suggesting higher pre-exercise endothelial NO synthesis in this group. Plasma creatinine levels tended to be higher in OMN compared to LOV and especially to VEG. Expectedly, men had higher plasma creatinine concentrations compared to women (106 ± 34.5 vs. 80.5 ± 17.2 µM, *p* < 0.001).

Regarding the pre-exercise plasma amino acids, VEG had the highest concentrations of Gly (*p*_omn_ = 0.002, *p*_LOV_ = 0.002), Asp+Asn (*p*_LOV_ < 0.001, *p*_OMN-LOV_ = 0.001) and Arg (*p*_LOV_ = 0.004, *p*_OMN-LOV_ = 0.011). In contrast, OMN had the highest basal levels of Leu+Ile (*p*_LOV_ = 0.033), guanidinoacetate (*p*_LOV_ = 0.001, *p*_LOV-VEG_ = 0.006), Glu+Gln (*p* = 0.041), Lys (*p*_LOV_ < 0.001, *p*_LOV-VEG_ = 0.006), homoarginine (*p*_LOV_ < 0.001), and Trp (*p*_LOV_ < 0.001, *p*_VEG_ = 0.005) before exercise test. The global arginine bioavailability ratio was lowest in LOV compared to the other groups (*p*_OMN_ = 0.005, *p*_VEG_ < 0.001). These observations indicate considerable differences in the plasma amino acids profile in the study groups pre-exercise.

### 3.2. Exercise-Induced Changes

MDA plasma concentrations increased post-exercise in all groups, yet statistically significant increases were observed in the LOV and VEG groups. No significant group-differences in changes of MDA were observed. Plasma nitrate, nitrite and creatinine did not change statistically significantly post-exercise. These observations suggest exercise-induced elevation of oxidative stress in the LOV and VEG groups, but no changes in NO metabolism and kidney function.

With respect to the plasma amino acids and their metabolites, the concentration of the majority decreased statistically significantly upon exercise, suggesting exercise-induced consumption of these amino acids. Yet, considerable increases were seen for Ala: +28% in OMN, +21% in LOV and +28% in VEG. The plasma concentrations of Thr, Gly and Val decreased upon exercise in OMN and LOV. Upon exercise, the plasma concentration of Ser increased (+16%) in OMN, but decreased in LOV (−17%) and VEG (−19%). The highest post-exercise increase was observed for the plasma concentration of Trp (+33%) in the OMN group. Analogous to Ser, the greatest post-exercise decrease was observed for the plasma concentration of Trp (−25%) in the LOV group.

Upon exercise, the plasma concentration of Sar, which is the *N*-methylated glycine, increased in the LOV (+11%) and VEG (+17%) groups, but decreased slightly in the OMN group (−1.6%). The plasma concentration of Asp+Asn decreased significantly in LOV and VEG after exercise. Exercise-induced changes were observed for the plasma concentrations of Orn + Cit and Tyr in OMN (highest) and in VEG (lowest changes; *p* = 0.045 and *p* = 0.004, respectively). The plasma concentrations of Phe and Tyr decreased upon exercise to almost the same degree in OMN and LOV, but did decrease only to a small extent in VEG.

The plasma Lys concentration changed upon exercise in OMN (decrease) and LOV (increase); these changes differed significantly between OMN and VEG (*p* = 0.001). The plasma Arg concentration decreased statistically significantly upon exercise in OMN and LOV. Yet, this did not result in significant changes of the global arginine bioavailability ratio in all three groups. The plasma concentrations of the Arg metabolites homoarginine and guanidinoacetate did not change significantly in all groups post-exercise. The guanidinoacetate/homoarginine molar ratio increased in all groups upon exercise, yet the increase (+12%) was only in LOV significant. The plasma Glu+Gln concentration did not change upon exercise in all groups. Exercise-induced significant decreases in the plasma concentration of Leu+Ile were found in the OMN (−17%) and LOV (−9.4%) groups.

Although all three groups showed lower pre-exercise concentrations of Pro, the largest difference was observed in OMN, who differed significantly from VEG (*p* = 0.015).

Regarding homoarginine, the highest exercise-induced reduction was obtained in OMN, who differed significantly from VEG (*p* = 0.010); the latter had elevated levels after exercise.

The above mentioned observations indicate that plasma amino acids are differently managed by the groups during exercise.

### 3.3. Correlations

In all three groups, plasma nitrate concentration pre-exercise correlated with plasma nitrate concentration post-exercise (*r*_omn_ = 0.808, *p*_omn_ < 0.001; *r*_LOV_ = 0.704, *p*_LOV_ < 0.001; *r*_VEG_ = 0.639, *p*_VEG_ = 0.001). Plasma nitrite (*r* = 0.470, *p* = 0.018) and creatinine (*r* = 0.723, *p* <0.001) concentrations pre- and post-exercise correlated only in OMN. Plasma MDA concentrations pre- and post-exercise correlated only in VEG (*r* = 0.518, *p* = 0.011). The plasma concentrations of the following amino acids correlated in all three groups pre- and post-exercise: Ala, Thr, Gly, Val, Sar, Leu+Ile, Asp+Asn, Pro, Met, Glu+Gln, Orn + Cit, Phe, Tyr, Lys, Arg, and homoarginine (Appendix A). These findings suggest that exercise changes distinctly different lipid peroxidation, NO metabolism and plasma amino acid profile in the groups.

Correlation data of oxidative stress (MDA) and NO metabolism (nitrite and nitrate) with plasma amino acids are reported in Table 4. We found inverse correlations for MDA with guanidinoacetate pre-exercise and with Trp post-exercise. The inverse correlation of guanidinoacetate with both, MDA and nitrite, may suggest that guanidinoacetate is consumed during lipid peroxidation and endothelial NO synthesis. All significant correlations of nitrate were positive. Except for Trp pre-exercise all significant correlations of nitrite were negative. The global arginine bioavailability ratio correlated positively with MDA only post-exercise.

### 3.4. Dietary Intake

In the whole study population, plasma MDA concentration pre-exercise (MDA_pre_) was positively associated with vitamin E intake (*r* = 0.258, *p* = 0.027). Plasma MDA concentration post-exercise (MDA_post_) was positively associated with α-linolenic acid intake (*r* = 0.271, *p* = 0.020), as well as with percentage of polyunsaturated fatty acids (*r* = 0.258, *p* = 0.027). Correlations between dietary intake (see Table 2) and plasma concentrations of amino acids pre- and post-exercise are summarized in Table 5. All found correlations were positive. The highest correlations were found for Val and Leu+Ile both pre- and post-exercise.

### 3.5. Associations with Exercise Capacity

MDA_post_ was positively correlated with maximum blood lactate concentration [Lac_max_] (*r* = 0.245, *p* = 0.040). A positive correlation was found between the plasma nitrate concentration post-exercise and maximum blood glucose concentration [Glc_max_] (*r* = 0.244, *p* = 0.040). These positive correlations suggest that lipid peroxidation and NO metabolism are associated with glucose metabolism. The plasma guanidinoacetate concentration pre-exercise was inversely correlated with body weight related maximum power output (*r* = −0.261, *p* = 0.030). Further, guanidinoacetate plasma concentrations pre- and post-exercise were inversely correlated with [Glc_max_] (pre: *r* = −0.304, *p* = 0.012; post: *r* = −0.264, *p* = 0.040) and with the lean body mass-related maximum power output P_maxLBM_ (pre: *r* = −0.321, *p* = 0.007; post: *r* = −0.299, *p* = 0.017). In contrast to lipid peroxidation and NO metabolism, guanidinoacetate seems to be utilized for energy generation, presumably serving as creatine precursor in skeletal muscles.

## 4. Discussion

### 4.1. Pre-Exercise Status of Plasma Oxidative Stress, NO Metabolism and Amino Acids Profile

Plant-based nutrition is constantly gaining popularity. Even though more and more athletes pursue vegetarianism or veganism, exercise-induced oxidative stress and metabolism are not well understood. To our knowledge, the sole reported work on oxidative stress among recreational athletes with vegetarian pattern is from Vanacore and colleagues [16]. This group found higher levels of thiobarbituric acid-reactive substances and lower nitrite levels after incubation of untreated and H_2_O_2_-treated H9c2 cells with serum in vegans, vegetarians and omnivores. Vanacore et al. concluded that restrictive vegan diet had minor antioxidant capacity compared to the other diets. Yet, the major limitation of that study is its in vitro nature. Based on a study on previously untrained and pre-trained healthy young men, Ristow and colleagues concluded that exercise-induced oxidative stress is responsible for health-promoting effects, such as insulin sensitization [7]. Basically based on the concept of mitohormesis it was assumed that supplementation with antioxidants may preclude health-promoting effects of exercise in humans [7]. It could, therefore, be assumed that higher circulating concentrations of MDA, the prominent thiobarbituric acid-reactive substance and measure of lipid peroxidation, in vegans may be associated with health effects and could explain the comparably low prevalence of diabetes in subjects who practice a plant-based diet [11].

The study we report here is the first investigation examining changes in physical exercise-induced oxidative stress measured as plasma MDA, and in NO and amino acids metabolism in three groups of recreational runners, i.e., omnivorous (OMN), lacto-ovo-vegetarian (LOV) and vegan (VEG). The study participants did not differ in age and gender distribution, had closely comparable BMI values and no statistically different training frequency (Table 1). The highest dietary intake of the antioxidant vitamins C and E was recorded in the VEG group (Table 2). The daily dietary intake of α-linolenic acid intake and polyunsaturated fatty acids in terms of percentage energy was numerically, but not statistically significantly higher in the VEG group. Pre-exercise, the subjects of the three groups of our study had closely comparable plasma MDA concentrations. Based on the currently generally accepted assumption that higher circulating MDA concentrations indicate higher oxidative stress, the results of our study suggest that healthy young VEG subjects are not stronger prevented from exercise-induced oxidative stress by the reportedly higher intake of antioxidants.

In our study, the pre-exercise mean plasma nitrate and nitrite concentrations were considerably higher in the subjects of the LOV group (+23%) and of the VEG group (+41%) compared to those of the OMN group. This is most likely due to the higher nitrate content of the vegetarian dietary patterns [33]. Methionine (Met) is considered to possess antioxidative properties, in part due to its sulphur atom. Reportedly, the subjects of the VEG group had a lower daily dietary intake of Met (Table 2). However, the pre-exercise concentrations of Met measured in the plasma samples of the VEG subjects were closely comparable to those of the OMN and LOV groups (Table 3). Unfortunately, our gas chromatographic-mass spectrometric method for amino acids does not allow measurement in plasma of the Met-relatives, each a thiol group-containing cysteine and glutathione [34]. With respect to the plasma concentrations of the other amino acids and their metabolites, we found rather moderate differences among the three groups pre-exercise (Table 3). The plasma concentrations of Gly, Asp+Asn and Arg were highest in the VEG group, in contrast to the energy-related Leu+Ile, and to Trp, which were the lowest concentrations in the VEG group. The highest plasma levels of Leu+Ile, guanidinoacetate, Glu+Gln, Lys, homoarginine, and Trp were measured in the OMN group. The plasma levels of Asp+Asn, guanidinoacetate, Glu+Gln, Lys, Arg and homoarginine were lower in the LOV group compared to the VEG group, which partly agrees with findings of the EPIC Oxford cohort [25]. For Val, Leu+Ile, Phe, Tyr, and Lys we found correlations to the dietary intake, which is largely consistent with previous findings [25]. Interestingly, the global arginine bioavailability ratio, i.e., the concentration ratio of plasma Arg to the sum of Cit and Orn concentration, did not differ between the groups pre-exercise suggesting no substantial differences in Arg-involving pathways including the urea cycle and the L-Arg/NO pathway.

The subjects of the VEG group had borderline lower plasma creatinine concentrations, presumably due to the creatine-poor or creatine-free vegan diet.

### 4.2. Exercise-Induced Effects on Oxidative Stress, NO Metabolism and Amino Acids Profile

We observed multiple and complex effects of physical exercise on oxidative stress, NO metabolism and amino acids profiling in the plasma of the subjects of the three groups.

Exercise led to increases in plasma MDA concentration, with the highest percentage increases being seen in the LOV (+24%) and VEG (+15%) groups, suggesting higher exercise-induced elevation of oxidative stress in these groups compared to the OMN group. MDA is produced by free radicals and enzyme-catalyzed lipid peroxidation of PUFAs including arachidonic acid and ALA [1]. Previous results [1] suggest that the higher dietary intake of ALA and PUFAs may have, at least in part, contributed to the higher plasma MDA concentrations measured in the LOV and VEG groups.

Exercise is generally considered to increase NO formation in the vasculature due to shear force-induced elevation of endothelial NO synthase. In our study, physical exercise did not cause statistically significant changes in plasma nitrate and nitrite concentrations, suggesting no appreciable effects of exercise on NO synthesis and metabolism. The lack of appreciable changes in plasma nitrite concentrations in the three groups suggests that the exercise (incremental stress test on a bicycle ergometer) did not alter the metabolism (e.g., reduction of nitrate to nitrite) or excretion/reabsorption of nitrite. The latter is supported by the lack of changes in plasma creatinine concentrations in all groups, suggesting unaffected renal function in terms of glomerular filtration rate.

The plasma concentration of the majority of the amino acids decreased upon exercise. This could be explained by catabolic processes such as transamination, oxidation, and gluconeogenesis, rather than due to changes in their renal filtration and reabsorption. It is worth mentioning that in gluconeogenesis up to 19% of energy can be obtained from Ala [21,35]. This concurs with the increase in the plasma concentration of Ala seen in all groups of our study post-exercise. Ala is likely to be provided by other organs and cells via cross-talk [33].

Regarding Pro, significant decreases were observed from pre- to post-exercise in OMN and LOV. These findings are consistent with the literature and may reflect the transformation to glutamate (Glu) and further to glutamine (Gln) [19,21]. Pro provides the highest energy capacity of non-essential amino acids (33 mol ATP per mol Pro). Interestingly, while Sar decreased post-exercise in OMN, an increase in LOV and VEG was observed. Since the latter groups consumed lower Met compared to OMN, these results suggest that LOV and VEG compensated the energy supply via Sar.

Guanidinoacetate is the direct precursor of creatine, which, in turn, is the direct precursor of the energy-related creatine phosphate (i.e., phospho-creatine). Creatine is present in large amounts in meat and fish, while vegetarian and vegan food contain very low amounts of creatine. The plasma concentration of guanidinoacetate did not increase significantly in the LOV (+10%) and VEG (+12%) groups upon exercise, while it decreased non-significantly (−13%) in the OMN group. Although not statistically significant, these changes may suggest that exercise induced consumption of guanidinoacetate in the OMN group to form creatine mainly in the kidney, liver and pancreas, on the one hand, and elevated enzymatic synthesis of guanidinoacetate in the LOV and VEG groups, on the other hand.

In fact, guanidinoacetate is produced from l-arginine and glycine by the catalytic action of l-arginine:glycine amidinotransferase (AGAT), which is abundantly expressed in the kidney [36]. AGAT also catalyses the synthesis of l-homoarginine, the methylene homolog of l-arginine, from l-arginine and l-lysine (Lys) [36]. Exercise induced similar, yet non-significant changes in the plasma concentrations of homoarginine: −11% in the OMN group, +1.2% in the LOV group and +5% in the VEG group. The relatively uniform changes in guanidinoacetate and homoarginine plasma concentrations in all groups may suggest that exercise increased the AGAT activity in the LOV group and more strongly in the VEG group. Presumably, this is required to come up to the higher demand on energy in skeletal muscles of the LOV and VEG runners, who are very likely to intake less creatine by their diets compared to the OMN runners.

### 4.3. Associations with Exercise Capacity

As previously reported, the cohorts of the present study were adequately supplied with iron, vitamin B_12_ and vitamin D [37], and had comparable maximum power output levels (P_maxBW_) (OMN: 4.15 ± 0.48 Watt/kg, LOV: 4.20 ± 0.47 Watt/kg, VEG: 4.16 ± 0.55 Watt/kg; *p* = 0.917) as well as lactate concentrations (OMN: 11.3 ± 2.19 mM, LOV: 11.0 ± 2.59 mM, VEG: 11.9 ± 1.98 mM; *p* = 0.648) [32]. The correlations of the post-exercise plasma concentrations of MDA, nitrate and guanidinoacetate with the blood concentrations of lactate and glucose suggest that oxidative stress, NO metabolism and AGAT activity/expression are associated with lactate and glucose metabolism, albeit in opposite direction: positive for MDA and nitrate, and inverse for guanidinoacetate. However, oxidative stress was not associated with exercise capacity in the form of body weight- and lean body mass-related maximum power output, suggesting that high levels of oxidative stress may not affect exercise capacity. But the causality and the effect of higher plasma MDA levels remain to be elucidated. We found a correlation of nitrate_post_ and [Glc_max_], which may reflect NO-dependent regulation of glucose uptake in skeletal muscle [38]. Thus, local NO increase during exercise might have favorable effects on glucose uptake. Previous studies found positive associations of a high dietary intake of nitrate-rich foods, such as beetroot, and athletic performance [39]. Bacterial nitrate reductase, which is present in mouth and gut flora, converts orally taken inorganic nitrate into nitrite, which in turn can be further converted to NO under certain conditions such as hypoxia. It would be interesting to examine oxygen consumption in future studies. The AGAT-catalysed synthesis of guanidinoacetate is the rate-limiting step in creatine production. The consistently inverse correlation of plasma guanidinoacetate concentration with body weight- and lean body mass-related maximum power output may suggest that the AGAT activity/expression may be a limiting factor in physical exercise.

### 4.4. Strengths and Limitations of the Study

A strength of our study is the use of fully validated and clinically proven gas chromatographic-mass spectrometric methods for the measurement of MDA, nitrate, nitrite and the majority of amino acids in plasma samples. Potential limitations of our study may be the relatively small number of participants included in the three groups and the methodological inability to discriminate between Orn and Cit, and to measure cysteine and glutathione, two important endogenous antioxidants. Nevertheless, the relatively constancy of the global arginine bioavailability ratio in the groups pre-exercise and post-exercise suggest that major Arg-involving pathways are not affected by diet and physical exercise. Our study may also be limited by the fact that the analysis of the above mentioned parameters may have been affected by pre-analytical factors. The nutrient intake via a 24 h dietary recall reflects the actual but not necessarily the usual daily consumption; deviations due to subjective estimation of the stated amounts of nutrients are possible. We used nutrition-specific software for calculations which cannot accurately estimate the dietary intake of each of the amino acids examined in plasma.

### 4.5. Future Research Directions

It would be interesting to investigate the effects of dietary pattern and exercise intensity and endurance on oxidative stress, NO and amino acids metabolism in the general population and in other sport disciplines.

## 5. Conclusions

Pre-exercise, OMN, LOV and VEG recreational athletes had different states of oxidative stress, when measured as plasma MDA concentration, different pre-exercise plasma amino acid profiles, but comparable NO metabolism. The greatest concentration of plasma guanidinoacetate, the direct precursor of the energy-related creatine, was highest in the OMN athletes. In all groups, physical exercise induced elevation of oxidative stress, but caused different changes in amino acids metabolism. The plasma concentration of Ala, the most important amino acid in gluconeogenesis, increased uniformly in all groups. The changes observed are related in part to glucose metabolism and in part to the different management of creatine homeostasis. Guanidinoacetate is differently utilized by OMN, LOV and VEG.

## Figures and Tables

**Table 1 nutrients-11-01875-t001:** Characterization of the study populations.

	OMN (*n* = 25)	LOV (*n* = 25)	VEG (*n* = 23)	*p* Value
Age (years)	27.2 ± 4.1	27.6 ± 4.4	27.3 ± 4.3	0.917^a^
Gender (*n*, males/females)	10/15	10/15	8/15	0.913^b^
BMI (kg/m^2^)	22.3 ± 1.8	21.5 ± 1.9	21.9 ± 2.2	0.412^a^
Weekly training frequency	3.0 ± 0.9	3.2 ± 0.9	3.0 ± 0.9	0.757^a^
Weekly running (h)	2.7 ± 1.1	3.3 ± 1.3	2.7 ± 1.4	0.122^a^

OMN = omnivores; LOV = lacto-ovo-vegetarians; VEG = vegans. Data are presented as mean ± SD. ^a^ Kruskal-Wallis test, ^b^ Chi square test.

**Table 2 nutrients-11-01875-t002:** Dietary intake of antioxidants and amino acids (mean ± SD) by diet group.

Parameters	OMN	*p* Value OMN vs. LOV	LOV	*p* Value LOV vs. VEG	VEG	*p* Value OMN vs. VEG	*p* Value
Antioxidants (mg)							
Vitamin C	143 ± 153	n.s.	148 ± 145	n.s.	218 ± 138	n.s.	0.037
Vitamin E	12.0 ± 6.36	n.s.	13.1 ± 11.0	0.040	19.7 ± 12.0	n.s.	0.021
Amino acids (g)							
Arg	4.57 ± 2.60	-	3.75 ± 2.21	-	4.53 ± 2.72	-	0.442
Thr	3.60 ± 2.40	-	2.95 ± 1.86	-	2.47 ± 1.25	-	0.185
Val	4.93 ± 3.04	-	4.38 ± 2.73	-	3.51 ± 1.87	-	0.213
Leu+Ile	11.4 ± 7.10	-	10.2 ± 6.60	-	8.05 ± 4.51	-	0.200
Met	1.90 ± 1.24	n.s.	1.56 ± 1.17	n.s.	0.99 ± 0.61	0.004	0.005
Phe	4.02 ± 2.23	-	3.68 ± 2.24	-	3.29 ± 1.80	-	0.507
Tyr	3.27 ± 1.89	-	3.04 ± 2.08	-	2.29 ± 1.33	-	0.134
Lys	5.64 ± 3.85	-	4.57 ± 3.21	-	3.36 ± 1.97	-	0.062
Trp	1.07 ± 0.64	-	0.91 ± 0.54	-	0.81 ± 0.39	-	0.346
Fatty acids							
ALA (g)	1.43 ± 1.58	-	1.85 ± 2.27	-	2.16 ± 1.65	-	0.115
PUFA (EN%)	4.87 ± 2.30	-	4.76 ± 2.34	-	6.26 ± 2.94	-	0.085

ALA, alpha-linolenic acid; PUFA, polyunsaturated fatty acids; EN% = energy percent; n.s., not significant. Statistical analysis was performed with Kruskal-Wallis test to examine group differences. Post-hoc test was conducted for statistically significant differences.

**Table 3 nutrients-11-01875-t003:** Plasma concentrations (mean ± SD; mM for creatinine; µM for the other analytes) and mean percentage changes (Δ, %) of the biochemical parameters of the three groups pre-exercise (pre) and post-exercise (post).

Parameters	OMN			LOV			VEG		
	Pre	Post	Δ (%)	Pre	Post	Δ (%)	Pre	Post	Δ (%)
Oxidative Stress/NO Metabolism	(*n* = 25)			(*n* = 25)			(*n* = 23)		
MDA	0.52 ± 0.09	0.56 ± 0.10	+9.2	0.50 ± 0.07	0.62 ± 0.15 †	+24	0.57 ± 0.13	0.68 ± 0.15 †	+15
Nitrate	70.7 ± 15.9	70.5 ± 17.8	−0.3	91.9 ± 43.8	94.9 ± 50.8	+3.3	120 ± 146	102 ± 41.9	−5.7
Nitrite	1.93 ± 0.26	1.85 ± 0.21	−4.4	2.67 ± 0.61	2.56 ± 0.53	−4.2	2.50 ± 1.09	2.36 ± 0.42	−18
Kidney function									
Creatinine	94.1 ± 20.7	93.4 ± 17.3	−0.7	90.0 ± 32.8	92.4 ± 40.2	+2.7	86.4 ± 37.6	84.7 ± 20.7	−2.1
Amino acids	(*n* = 24)			(*n* = 25)			(*n* = 22)		
Ala	411 ± 100	530 ± 112 ‡	+28	400 ± 85.9	485 ± 111 ‡	+21	439 ± 104	557 ± 117 ‡	+28
Thr	194 ± 75.0	167 ± 63.4 ‡	−15	161 ± 47.9	148 ± 66.6 †	−8.0	158 ± 41.7	157 ± 62.9*	−0.1
Gly	249 ± 68.4	231 ± 57.6 *	−7.4	248 ± 77.5	227 ± 63.8 †	−8.3	324 ± 79.5	312 ± 70.8	−4.2
Val	373 ± 150	332 ± 104 †	−12	332 ± 116	302 ± 99.9 *	−8.9	298 ± 106	284 ± 75.8	−4.2
Ser	204 ± 152	235 ± 220	+16	168 ± 59.9	147 ± 78.6 *	−17	185 ± 91.5	150 ± 22.3 *	−19
Sar	2.24 ± 0.99	2.22 ± 1.00	−1.6	1.84 ± 0.55	2.05 ± 0.64 †	+11	1.66 ± 0.46	1.95 ± 0.51 †	+17
Leu +Ile	300 ± 125	251 ± 76.5 †	−17	231 ± 102	209 ± 74.7 *	−9.4	227 ± 75.9	254 ± 127	+12
GAA	3.79 ± 1.24	3.25 ± 0.89	−13	2.63 ± 0.64	3.01 ± 0.94	+9.9	3.42 ± 0.74	3.88 ± 1.02	+12
Asp+Asn	104 ± 28.9	94.7 ± 32.2	−9.7	75.8 ± 17.0	66.1 ± 18.7 †	−13	107 ± 22.9	97.0 ± 17.3 *	−9
Pro	213 ± 75.3	188 ± 59.9 †	−12	220 ± 73.2	199 ± 66.6 †	−9.6	214 ± 59.1	209 ± 67.9	−0.8
Met	65.5 ± 13.0	62.6 ± 12.7 *	−4.4	60.2 ± 7.03	57.7 ± 7.88	−4.2	65.5 ± 6.94	65.4 ± 6.78	+0.2
Glu+Gln	802 ± 236	787 ± 235	−1.6	682 ± 106	654 ± 116	−4.1	766 ± 107	773 ± 117	+1.6
Orn+Cit	59.8 ± 21.2	50.1 ± 15.8 ‡	−17	55.1 ± 13.7	48.0 ± 12.1 ‡	−13	59.0 ± 19.6	56.2 ± 18.3	−3.8
Phe	80.6 ± 20.6	72.0 ± 17.5 †	−12	68.1 ± 17.4	61.8 ± 13.9 †	−9.2	73.7 ± 16.6	70.7 ± 12.9	−3.3
Tyr	75.3 ± 34.1	66.2 ± 25.3 †	−13	65.9 ± 25.4	60.6 ± 21.2 †	−9.4	58.3 ± 18.0	58.1 ± 15.8	−0.4
Lys	206 ± 69.1	184 ± 51.0 †	−12	109 ± 37.5	120 ± 39.9 *	+9.3	153 ± 47.2	149 ± 33.4	−1.9
Arg	91.8 ± 29.7	77.4 ± 21.0 †	−12	69.1 ± 19.1	66.5 ± 18.2 *	−6.0	93.5 ± 25.9	90.3 ± 23.6	−2.9
hArg	1.89 ± 0.93	1.64 ± 0.93	−11	1.09 ± 0.39	1.10 ± 0.38	+1.2	1.51 ± 0.81	1.58 ± 0.75	+5
Trp	34.5 ± 11.6	45.8 ± 68.8	+33	22.9 ± 9.28	17.2 ± 5.43 ‡	−25	22.7 ± 4.70	18.6 ± 5.45	−18
GAA/hArg	2.26 ± 0.91	2.53 ± 1.22	+11	2.62 ± 1.25	2.98 ± 1.49 *	+12	2.79 ± 1.34	2.86 ± 1.21	+2.5
GABR	1.57 ± 0.33	1.58 ± 0.30	+6.3	1.27 ± 0.27	1.41 ± 0.31	+17	1.65 ± 0.34	1.67 ± 0.32	+1.9

+ indicates increase, − indicates decrease. Asterisks indicate statistical differences (* *p* < 0.05; † *p* < 0.01; ‡ *p* < 0.001). GAA, guanidinoacetate; hArg, homoarginine; GABR, global arginine bioavailability ratio.

**Table 4 nutrients-11-01875-t004:** Spearman correlation coefficients (*r*) and *p* values of circulating amino acids with MDA, nitrate and nitrite in the combined groups.

Parameter at the Respective Time	MDA (*r*, *p*)		Nitrate (*r*, *p*)		Nitrite (*r*, *p*)	
Pre-Exercise						
Ala	n.s.	n.s.	0.290	0.013	n.s.	n.s.
Gly	n.s.	n.s.	0.391	0.001	n.s.	n.s.
GAA	−0.247	0.041	n.s.	n.s.	−0.251	0.038
Lys	n.s.	n.s.	n.s.	n.s.	−0.361	0.002
hArg	n.s.	n.s.	n.s.	n.s.	−0.469	<0.001
Trp	n.s.	n.s.	n.s.	n.s.	0.237	0.048
GABR	n.s.	n.s.	n.s.	n.s.	n.s.	n.s.
Post-exercise						
Gly	n.s.	n.s.	0.495	<0.001	n.s.	n.s.
Met	n.s.	n.s.	0.242	0.040	n.s.	n.s.
Orn+Cit	n.s.	n.s.	0.264	0.025	n.s.	n.s.
Ser	n.s.	n.s.	n.s.	n.s.	−0.274	0.030
Leu+Ile	n.s.	n.s.	n.s.	n.s.	−0.286	0.015
Asp+Asn	n.s.	n.s.	n.s.	n.s.	−0.308	0.008
Lys	n.s.	n.s.	n.s.	n.s.	−0.442	<0.001
Trp	−0.246	0.045	n.s.	n.s.	−0.299	0.014
GABR	0.275	0.029	n.s.	n.s.	n.s.	n.s.

GAA, guanidinoacetate; hArg, homoarginine; GABR, Global Arg bioavailability ratio; n.s., not significant.

**Table 5 nutrients-11-01875-t005:** Spearman correlation coefficients (*r*) and *p* values of dietary intake and plasma concentrations of amino acids in the whole cohort.

	Pre-Exercise		Post-Exercise	
Amino Acids	*r*	*p*	*r*	*p*
Arg	0.317	0.007	0.282	0.025
Thr	0.256	0.030	0.190	0.109
Val	0.560	<0.001	0.571	<0.001
Leu+Ile	0.459	<0.001	0.421	<0.001
Met	0.157	0.188	0.145	0.224
Phe	0.351	0.003	0.321	0.006
Tyr	0.341	0.003	0.380	0.001
Lys	0.335	0.004	0.342	0.003
Trp	0.129	0.286	0.324	0.007

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
