# Peer review of "Exercise-Induced Oxidative Stress, Nitric Oxide and Plasma Amino Acid Profile in Recreational Runners with Vegetarian and Non-Vegetarian Dietary Patterns"

_nutrients, 2019, doi:10.3390/nu11081875_

Round 1
Reviewer 1 Report
There are considerable issues with the paper. The grouping of certain amino acids, especially citrulline and ornithine is of concern as these two reflect quite different changes in urea cycle production and will hide the changes actually occurring. If one wants to evaluate the urea cycle and its relationship to the changes in Nitrate and nitrite one could use substrate product ratios to evaluate the various enzymes within the various cycles and pathways to assess the data. If one wants to quote dietary intakes one should also measure them in the serum to properly evaluate the changes.
Author Response
Point-by-point response to the comments made by the reviewers
We thank the reviewer for the evaluation of our paper and the suggestions made to improve it.
Reviewer´s comments: Reviewer 1
A) Responses concerning the general evaluation of the paper (Table)
1) We improved the Introduction which includes all relevant references
2) The study design was improved. A graphic is now reported in the Supplement as Figure S1.
3) We improved all the methods
4) The results are now clearly presented. Please consider that their presentation is likely to be dependent on Journal’s software.
5) We improved the conclusions which are now better supported by the results.
There are considerable issues with the paper. The grouping of certain amino acids, especially citrulline and ornithine is of concern as these two reflect quite different changes in urea cycle production and will hide the changes actually occurring. If one wants to evaluate the urea cycle and its relationship to the changes in Nitrate and nitrite one could use substrate product ratios to evaluate the various enzymes within the various cycles and pathways to assess the data. If one wants to quote dietary intakes one should also measure them in the serum to properly evaluate the changes.
B) Response concerning specific evaluation of the paper (Comments)
All analytes were quantitated in the plasma samples of our study by GC-MS. The GC-MS method for plasma/serum amino acids has been thoroughly validated and reported in previous work (Hanff et al. 2019). In that paper we also critically discussed the fact that the GC-MS method cannot discriminate between citrulline and ornithine and therefore their sum has to be calculated and used. In the present paper we also addressed this issue.
We agree that the availability of individual concentrations for citrulline and ornithine is likely to have provided additional information.
Using the concentrations of Arg and Cit+Orn we calculated the global arginine bioavailability ratio (GABR) by dividing the Arg concentration ([Arg]) by the sum concentration of Cit and Orn [Cit+Orn]:
GABR = [Arg]/[Cit+Orn]
We calculated GABR in all groups and performed statistical analyses. These new data are now included in the respective tables of the revised manuscript.
We also performed correlations between GABR with various parameters including nitrate and nitrate. These new data are also now presented in the respective tables of the revised manuscript.
The constancy of the GABR data suggests that Arg-involving pathways are similar in the three groups and do not change substantially upon exercise.
Reviewer 2 Report
The authors have presented a well-designed study of exercised-induced oxidative stress in OMN/LOV/VEG cohorts, with well-thought out discussion and concluding remarks.
There are only minor corrections:
Line 31: Insert "with" between "oxidative stress" and "no".
Line 59-60: define H9c2 and H-H9c2 cells, something along the lines of: " the rat cardiac myoblastic cell line, H9c2". Also here, I'm not sure what the "H" in "H-H9c2" means.
It is discussed that vegans have lower levels of Met, which is an interesting result. From my limited understanding, Met --> cysteine --> glutathione. With GSH being one of the major antioxidants in the body, would it be possible to go back (if not already done) and measure GSH levels in the participant's plasma, to perhaps correlate the Met result with why the VEG cohort have greater oxidative stress follwoing exercise?
In Table 2: P value column (far right), check VitC/VitE P values should not be under the heading "P value OMN vs VEG".
Author Response
Point-by-point response to the comments made by the reviewers
We thank the reviewer for the evaluation of our paper and the suggestions made to improve it.
Reviewer´s comments: Reviewer 2
Comments and Suggestions for Authors
The authors have presented a well-designed study of exercised-induced oxidative stress in OMN/LOV/VEG cohorts, with well-thought out discussion and concluding remarks.
There are only minor corrections:
Line 31: Insert "with" between "oxidative stress" and "no".
Response: We have inserted “with” between "oxidative stress" and "no".
Line 59-60: define H9c2 and H-H9c2 cells, something along the lines of: " the rat cardiac myoblastic cell line, H9c2". Also here, I'm not sure what the "H" in "H-H9c2" means.
Response: We added a definition of the H9c2 cells in the manuscript.
It is discussed that vegans have lower levels of Met, which is an interesting result. From my limited understanding, Met --> cysteine --> glutathione. With GSH being one of the major antioxidants in the body, would it be possible to go back (if not already done) and measure GSH levels in the participant's plasma, to perhaps correlate the Met result with why the VEG cohort have greater oxidative stress follwoing exercise?
Response: Unfortunately, we have not measured Cys and GSH. Cys and especially GSH are very difficult to measure by GC-MS, the method we have used for all other analytes.
Because we did not add to the plasma samples substances to stabilize Cys and GSH, their measurement by alternative techniques (e.g., HPLC with fluorescence detection) is not meaningful for our already thawed plasma samples.
In Table 2: P value column (far right), check VitC/VitE P values should not be under the heading "P value OMN vs VEG".
Response: Probably there is a formatting problem, since we cannot see the error named by the reviewer.
Round 2
Reviewer 1 Report
Tracking comments have been made in the paper. There are important relationships identified in the results data which are not commented upon. e.g. MDA, Nitrite and Nitrate each correlate with different amino acid sets yet no analysis is found to assess this and no comment made. The best way to assess this is when you are writing the discussion and you see an unclear conclusion, relook at your data to see if it can give you the evidence. There are several instances in the discussion where the analysis of the data should be able to allow you to draw a conclusion or present evidence to better discuss the findings.
Most results section have the presentation of data but then fail to have concluding sentence or paragraph to allow the reader to understand the findings of each section.
The overuse of abbreviations will be confusing to most readers (including me) as they spend a lot of time looking back to find out what does it mean. This detracts from the readers ability to GET THE MESSAGE. One should limit the number of abbreviations to 3-4 at the most.
These are a few minor spelling errors e.g. "katabolic" should be "catabolic".
Author Response
We performed the following changes which are indicate in yellow in the manuscript.
1) Typos including “catabolic” were corrected.
2) We understand the restriction of abbreviations. We reduced the number of abbreviations to a minimum, especially in the Abstract. Yet, the abbreviations used for regular amino acids are actually no abbreviations. We cannot resign the use of the standard 3-characters names of amino acids, especially in the Tables.
3) We clearly improved the Results section. When presenting the data, we added additional comments, descriptions and possible explanations for the findings
4) We also improved the Discussion section, thereby avoiding major repetitions.
5) We also improved the Conclusion section.